# Mitochondrial DNA Variants and Common Diseases: A Mathematical Model for the Diversity of Age-Related mtDNA Mutations

**DOI:** 10.3390/cells8060608

**Published:** 2019-06-18

**Authors:** Huanzheng Li, Jesse Slone, Lin Fei, Taosheng Huang

**Affiliations:** 1Human Aging Research Institute, Nanchang University, Nanchang 330031, China; oceanuniversity@126.com; 2Wenzhou Key Laboratory of Birth Defects, Wenzhou Central Hospital, Wenzhou 325000, Zhejiang, China; 3Division of Human Genetics, Cincinnati Children’s Hospital Medical Center, 3333 Burnet Avenue, Cincinnati, OH 45229, USA; jesse.slone@cchmc.org; 4Division of Biostatistics and Epidemiology, Cincinnati Children’s Hospital Medical Center, 3333 Burnet Avenue, Cincinnati, OH 45229, USA; lin.fei@cchmc.org

**Keywords:** mitochondria, mitochondrial genetics, aging, cancer, diabetes, mutation

## Abstract

The mitochondrion is the only organelle in the human cell, besides the nucleus, with its own DNA (mtDNA). Since the mitochondrion is critical to the energy metabolism of the eukaryotic cell, it should be unsurprising, then, that a primary driver of cellular aging and related diseases is mtDNA instability over the life of an individual. The mutation rate of mammalian mtDNA is significantly higher than the mutation rate observed for nuclear DNA, due to the poor fidelity of DNA polymerase and the ROS-saturated environment present within the mitochondrion. In this review, we will discuss the current literature showing that mitochondrial dysfunction can contribute to age-related common diseases such as cancer, diabetes, and other commonly occurring diseases. We will then turn our attention to the likely role that mtDNA mutation plays in aging and senescence. Finally, we will use this context to develop a mathematical formula for estimating for the accumulation of somatic mtDNA mutations with age. This resulting model shows that almost 90% of non-proliferating cells would be expected to have at least 100 mutations per cell by the age of 70, and almost no cells would have fewer than 10 mutations, suggesting that mtDNA mutations may contribute significantly to many adult onset diseases.

## 1. Introduction

The mitochondrial organelle, a double-membraned organelle with its evolutionary origins in the eubacterial kingdom [1], is the central factor in the energy metabolism of the eukaryotic cell. It is responsible for the vast majority of the ATP (adenosine triphosphate) produced in the cell (~90% under normal circumstances), which it produces through oxidative phosphorylation (OXPHOS) by way of a multi-subunit complex called the electron transport chain (ETC) [2]. Most of the proteins required by the mitochondrion to both propagate itself as well as to maintain its vital biochemical functions are produced from genes located in the nuclear genome. These proteins are first produced in the cytoplasm before being imported into the mitochondrion, and thus they are classified as “nuclear” genes despite their functional role in the mitochondria.

However, nuclear genes are not the only genetic factors that determine the function of the mitochondrion. The mitochondrion also possesses its own small, circular genome that encodes key RNA and proteins required for OXPHOS. Despite being relatively small in length (~16.5 kb), this mitochondrial DNA (or “mtDNA”) is rich in content, containing 13 protein-coding, 22 tRNA, and 2 rRNA genes, all encoded via polycistronic (multigene) transcripts. The 13 protein-encoding genes are distributed between various respiratory chain components (specifically, complexes I, III, IV, and V). Only a small amount of noncoding DNA is found between these tightly spaced genes, with the D-loop (located from 16,028 to 577 bp) comprising the largest portion of noncoding mtDNA (and ~6% of the total mtDNA). The mitochondrion is the only human organelle that has independent DNA replication, transcription, and translation systems. Therefore, the 22 tRNA and 2 rRNA genes required for translation are also produced by mtDNA [3], so that these operations can be carried out directly within the organelle.

The mtDNA mutation rate depends on many factors, including the extent of oxidative stress and the fidelity of the mitochondrial DNA polymerase (POLG). The production of reactive oxygen species (ROS) is an inevitable outcome of the oxidative phosphorylation process that occurs within the mitochondrion, and these chemical byproducts are, by their nature, damaging to DNA. Thus, given its proximity to the source of ROS production, mtDNA experiences a high rate of ROS-induced mutation. ROS production is also increased by pre-existing mtDNA damage, excess calories, regional mtDNA genetic variations, and alterations in nDNA expression of stress response genes [4], creating a vicious cycle where increased ROS production can encourage the occurrence of even more ROS production over time. The mtDNA also has an exceptionally high mutation rate due to some unique features of the mitochondrion. For instance, somatic mtDNA damage usually exists in a state of “heteroplasmy,” meaning that mutant and wild-type mtDNAs coexist. Heteroplasmic mtDNA segregate randomly during cell division, allowing for shifts in heteroplasmy. Since human cells contain hundreds or even thousands of copies of mtDNA (with each cell containing different proportions of mutant and wild-type mtDNA), great variability in mtDNA occurs as humans age. This may contribute significantly to the variability of common diseases.

As one might predict, mutations in the mtDNA are a critical factor in the occurrence of mitochondrial disease. Since the first disease linked directly to an mtDNA mutation was discovered by Douglas Wallace in 1988 [5], more than 300 pathogenic mtDNA variants have been linked to various forms of primary mitochondrial dysfunction. This makes a thorough discussion of each of these variants (much less the thousands of variants in nuclear mitochondrial genes) beyond the scope of this review. Instead, for the purposes of this review, we will focus primarily on a discussion of mtDNA mutations, how they are acquired, and how they may lead to disease in patients with common disease conditions.

## 2. Examples of Nuclear Mitochondrial Genes with a Role in Common Diseases

The role of the mitochondria goes far beyond the simple production of ATP; mitochondria are also responsible for ROS production in the cells, as well as the regulation of calcium signaling, biosynthesis, iron homeostasis, and apoptosis [6,7,8,9]. It should be unsurprising, then, that mitochondrial dysfunction lies at the heart of many common disease etiologies. The underlying cause of these mitochondrial defects is usually some combination of oxidative stress and/or mutations to genes required for proper mitochondrial function (whether in the mtDNA or in the large assortment of nuclear genes required for mitochondrial function). Many of these mutations result in a compromised ETC, leading to both a reduction in OXPHOS (and thus, less ATP) as well as increased levels of ROS (which “leak” out as a result of the less efficient ETC). This, in turn, can further damage the mitochondria, leading to a vicious feedback loop that often results in apoptosis and cell loss, particularly in energy-intensive cells such as neurons. Other pathogenic mutations interfere with the critical processes of mitochondrial fusion/fission and mitophagy (the autophagic disposal of redundant or damage mitochondria), with potentially devastating consequences for the patient.

Here, before delving into pathologies related to mtDNA variants, we will highlight findings from the current literature on variants in mitochondrial-related nuclear genes and their associated diseases. By briefly summarizing these results, we hope to impart to the reader a better sense of how these nuclear genes intersect with common diseases.

### 2.1. Parkinson’s Disease

Parkinson’s disease (PD) is the second most common neurodegenerative disorder after Alzheimer’s disease. Patients with this disease can present with motor defects (such as resting tremor and myotonia) as well as non-motor defects such as depression, olfactory deficits, and insomnia [10]. The relationship between PD and mitochondrial-related nuclear gene dysfunction has been extensively described in the literature. In particular, the proteins PTEN-induced kinase 1 (PINK1) and Parkin (an E3 Ubiquitin Protein Ligase) play a central role in the pathology of PD, to the point that the latter protein was directly named after Parkinson’s disease. These two proteins act as powerful regulators of mitochondrial morphology and health by mediating the process of “mitophagy,” the autophagy-based pathway for recycling damaged or redundant mitochondria.

The mechanism by which PINK1 and Parkin induce mitophagy is relatively straightforward. Whenever a mitochondrial organelle exhibits any signs of compromised function, such as a drop in mitochondrial membrane potential, PINK1 (which is normally located along the inner mitochondrial membrane) will relocate to the outer mitochondrial membrane [11]. Here it recruits Parkin to the outer mitochondrial membrane, which it proceeds to activate via phosphorylation. This, in turn, activates Parkin’s ubiquitin ligase activity, leading to the ubiquitination of a variety of mitochondrial outer membrane proteins. This ubiquitination process is the decisive mark that ultimately dooms the mitochondria to destruction via mitophagy [12]. Mutations in either PINK1 or Parkin lead to PD by a variety of mechanisms related to the loss of mitochondrial quality control, including reduced OXPHOS, increased levels of ROS, altered calcium homeostasis, etc. [13,14,15,16]. In fact, mutations in any of the downstream proteins that help recruit the autophagosome to damaged mitochondria (such as NIPSNAP1 and NIPSNAP2) also lead to a parkinsonian phenotype that can be attributed directly to reduced mitophagy [17]. As with other mitochondrial diseases, mitochondrial fusion/fission dynamics can also play a role in the disease pathology. For example, work in *Drosophila* has shown that heterozygous mutations of the *Drp1* gene are lethal when combined with either *PINK1* or *Parkin* mutations, and that the various mitochondrial, physiological, and motor defects of *PINK1* and *Parkin* mutants can be mitigated by transgenic overexpression of the *Drp1* gene [18].

### 2.2. Alzheimer’s Disease

As mentioned in the previous section, Alzheimer’s disease (AD) is the most commonly occurring form of age-related dementia. It is characterized by a gradual decline in memory and cognitive functions, primarily due to neurodegeneration. Mitochondrial defects are often observed in the brain tissue of AD patients [3], perhaps due in part to the regulatory role that mitochondria play in controlling apoptotic cell death that drives such neurodegeneration. There may also be a more direct relationship between AD and mitochondrial dysfunction, as the formation of misfolded amyloid-β (Aβ) aggregates (which forms the central pathological mechanism of AD) has been shown to have a significant effect on the expression of ETC components and the overall function of the mitochondria [19,20], as well as the induction of mitochondrial apoptosis [3]. There is even evidence that mitochondrial defects may actually help induce the formation of Aβ aggregates [21], potentially placing it upstream of the latter in the overall mechanism of AD pathology.

### 2.3. Huntington Disease

Huntington disease (HD) is an inherited neurodegenerative disease characterized by movement disorders, cognitive decline and psychiatric disorders [9,22]. HD is also well-known for its relatively late onset of symptoms, which usually begins between the ages of 30 and 40. The direct molecular cause of the disease is a stretch of CAG repeats in the *huntingtin* gene (*HTT*), which is highly expressed in neurons. Expansion of these CAG repeats beyond the 6–35 copies normally present in wildtype alleles leads to a pathological expansion of the corresponding polyglutamine tract in in the HTT protein that causes the protein to form misfolded aggregates with toxic properties [23]. These toxic aggregates ultimately lead to neurodegeneration in the brain, particularly in the striatum.

What is particularly intriguing about these toxic HTT aggregates, however, is their potential effect on mitochondrial function. Several studies have shown that mutant HTT disrupts mitochondrial metabolism with downstream effects on calcium homeostasis and neuronal survival [24], and that it can also alter the localization and morphology of the mitochondria [25]. On the latter point, mutant HTT has been shown to raise the levels of mitochondrial fission proteins while lowering the levels of mitochondrial fusion proteins [26]. In particular, mutant HTT appears to increase the activity of the mitochondrial fission protein DRP1 (Dynamin-related protein 1), leading to increased mitochondrial fragmentation, defective anterograde mitochondrial transport, and degenerating synapses [27]. This relationship was further confirmed by using a specific inhibitor (P110-TAT) to block DRP1 activity in mouse and cell culture models of HD, which resulted in a reduction in the neurodegeneration, mitochondrial dysfunction, and mortality normally observed in these models [28]. Transgenic overexpression of the PINK1 protein in Drosophila has also been shown to counteract the detrimental effects of mutant HTT on mitochondrial function, ATP production, and neuronal health [29], suggesting that inducing mitophagy may also represent an effective means of treating HD.

### 2.4. Diabetes

Given its central role in energy metabolism and ROS production, the mitochondrion would be expected to be a major contributor to the development of insulin resistance (IR) and diabetes. This can be observed even at the basic level of mitochondrial fusion/fission dynamics, with levels of mitochondrial fusion proteins such as OPA1 (Optic Atrophy 1) and MFN2 (Mitofusin 2) clearly altered in tissues from diabetic patients [30]. In fact, insulin has been shown to stimulate both the expression of mitochondrial fusion proteins as well as overall mitochondrial function in rat cardiomyocytes [31], further strengthening the link between insulin signaling, diabetes, and mitochondria. A more direct relationship between OXPHOS disruption and diabetes has also come from the mouse model, where disruption of the CR6-interacting factor 1 protein, which is required for the proper synthesis and insertion of ETC proteins into the inner mitochondrial membrane) leads not only to decreased OXPHOS activity, but also to failure of the beta cells and diabetes [32]. A similar phenomenon of impaired OXPHOS subunit import and function has also been observed in adipose tissue from patients with type 2 diabetes [33]. Mitochondria compromised by insulin resistance and diabetes also appear to possess reduced activity in the mitochondrial β-oxidation pathway, as evidenced by lower levels of β-oxidation enzymes like ACADVL (Acyl-CoA dehydrogenase very long chain) and trifunctional enzyme subunit α observed in brain tissues from diabetic patients [34]. This results in a buildup of toxic lipid intermediates such as ceramides and diacylglycerol that create even more oxidative stress and cellular damage [35].

### 2.5. Cancer

Since the mid-20th century, it has been observed that cancer cells tend to rely almost exclusively on glycolysis rather than oxidative phosphorylation for their energy needs. This metabolic tendency is usually referred to as the “Warburg Effect,” after the researcher who initially described the phenomenon [36], and strongly implicates mitochondrial function (or lack thereof) in the oncogenic process. This exclusive reliance on glycolysis provides several advantages to the cancer cell and confers a greater ability to survive under hypoxic conditions, lowers ROS levels due to the absence of OXPHOS, and allows more substrates to be allocated to cell proliferation. All of these factors allow cancer cells to metastasize and aggressively outcompete normal cells. It has been shown that mutations of the mitochondrial-related genes are associated with cancer. For example, succinate dehydrogenase (SDH) is an enzyme involved in the citric acid cycle and the ETC, and mutations in various subunits of the SDH complex have been shown to cause several different forms of cancer [37,38,39].

### 2.6. Aging

The underlying causes of the aging process remain only partially understood, despite decades of research on the topic. While it is clear that most organisms experience a progressive loss of cellular and physiological function as they grow older, an intense debate remains on which factors contribute most to the phenomenon. One of the more popular theories regarding the underlying cause of aging is the “cellular aging theory,” which posits that each cell has a set number of cell divisions that it can undergo before it stops dividing altogether. This phenomenon is known as “cellular senescence”. In this model, this limiting number of cell divisions is determined by the length of the telomeres that occupy the ends of each chromosome, which become slightly shorter with each round of cell division. As this shortening occurs, the cells gradually lose the ability to protect the end of their chromosome, leading to a loss of chromosome stability that eventually results in a permanent arrest of the cell cycle (or, even worse, uncontrolled cell growth and cancer) [3]. A relationship between elevated levels of ROS and telomere loss has long been suspected, but only recently has a molecular mechanism been confirmed that directly links the two processes through chemical damage to guanine bases [40].

Studies have shown significant changes in the expression of both autophagy/mitophagy genes (LAMP-2A, HSC70, and PINK1) as well as the mitochondrial fission protein DRP1 [41] in aged mice. The latter protein is of particular interest, as transgenic expression of Drp1 in *Drosophila melanogaster* has been shown to dramatically prolong the lifespan of flies, while also improving mitochondrial morphology and the overall health of the flies [42]. This effect is speculated to be the result of increased mitophagy (since mitochondrial fission is known to induce mitophagy), suggesting that inducing mitochondrial fission may be an effective route to prolonging lifespan and improving health in the elderly. Thus, while many of the details remain to be worked out, it is clear that mitochondrial activity exerts a major influence on the aging process and should be considered as a major focus of future research in geriatric medicine.

## 3. Mitochondrial DNA in Common Diseases

### 3.1. Mitochondrial DNA Variants and Alzheimer’s Disease (AD)

Damage to mtDNA may be a major cause of abnormal ROS production in AD, but the mechanism by which such damage is incurred remains a subject of debate. While some have proposed that ROS-induced oxidative damage is itself a driver of these AD-associated mtDNA mutations, the mutations that are actually observed in brain tissues derived from AD patients are more consistent with mtDNA replication errors than with errors induced by oxidative damage [23]. This suggests that defects in the machinery that ensures accurate replication of mtDNA may by a major contributor in AD pathogenesis, and indeed, lower activity for at least one base-excision repair enzyme (mitochondrial uracil DNA glycosylase) has been demonstrated in brain tissues from AD patients relative to normal controls [37]. In this case, however, the primary effect of lower uracil DNA glycosylase activity seems to be to lower the mtDNA copy number in AD patient brain samples, rather than to alter the mtDNA mutagenesis rate itself. Regardless, the association between mtDNA damage and the occurrence of AD remains fairly strong.

Mitochondrial DNA is only inherited from the mother in humans [43], and thus all mtDNA alleles are passed down as a single unit (or haplotype) through the maternal line. This, in turn, leads to the existence of distinct human populations with a shared haplotype based on a shared ancestry through the maternal line. These shared haplotypes are referred to as “haplogroups”. It has been shown that both mitochondrial single-nucleotide polymorphisms (SNPs) and haplogroups affect AD susceptibility by influencing mitochondrial metabolism. This association was strongly confirmed by a systematic analysis of mtDNA from frontal cortex samples of AD and non-AD patients. The analysis showed a 63% increase in the number of mutations in the noncoding mtDNA control region of AD patients compared to non-AD control tissues, a difference that increased to 130% in patients over the age of 80 [44]. Other publications have also shown the association between AD and mtDNA variants in different contexts. Fesahat et al. found that mtDNA haplogroups H and U might constitute risk factors for AD in a Persian population, and that variations within these haplogroups may be involved in AD expression in combination with environmental exposures [45]. Maruszak et al. showed that variants in essential mitochondrial genes (both in the nuclear genome as well as in the mtDNA) may contribute to late-onset AD risk. For instance, the mtDNA haplogroup HV and H clusters were risk factors for late-onset AD, whereas haplogroup K reduced the high late-onset AD risk commonly found with the APOE4+ genotype [46]. Based on 500 late-onset AD patients from Asturias, Coto et al. showed that APOE-ε4 and haplogroup H (defined by nucleotide 7028C) were both significantly associated with disease [47]. Santoro et al. applied a high-resolution analysis to explore the link between mtDNA haplogroups and the occurrence of Alzheimer’s disease among individuals from central and northern Italy. They identified sub-haplogroup H5 as a potential risk factor independent from the APOE genotype. The H5a subgroup of mtDNA molecules (harboring the 4336 transition in the previously AD-associated tRNAGln gene) was about threefold more extensively represented in AD patients than in controls [48]. The MT-ATP6 gene with the B5-defining variant G8584A has also been shown to have significantly decreased mitochondrial function, suggesting haplogroup B5 confers genetic susceptibility to AD in Han Chinese and that this effect is most likely mediated by the ancient variant G8584A [49].

In general, it appears that the AD-associated mtDNA point mutations are mostly located in the protein-coding, tRNA, and D-loop regions. Members of haplogroup T in a European population (specifically, those with T14178C and A15244G variants) showed a greater risk of developing dementia (although not specifically AD per se) [50]. While Tanaka et al. argued that mtDNA polymorphisms cannot be the single main cause of AD, they found that T961C and T856G variants may elevate AD risk, and that rare mutations in the protein-coding region may have a protective effect on high-risk populations carrying the APOE-ε4 allele [51]. Krishnan et al. identified cytochrome c oxidase deficiency in the hippocampus of sporadic AD patients, which was ultimately caused by the age-related accumulation of high levels of mtDNA 4977 deletions [52].

Ultimately, mitochondrial dysfunction in AD causes changes in ROS and oxygen free radicals, which in turn cause apoptosis and characteristic clinical symptoms. This "mitochondrial cascade hypothesis" may help explain some of the pathological and genetic features of sporadic AD. Although some mutation sites or haplogroups are clearly linked to the occurrence of AD, further research is needed into the mechanism(s) and timing of mtDNA variations that are acquired with age. Future developments in this area of research may provide valuable insights into this devastating disease.

### 3.2. Mitochondrial DNA Variants and Parkinson’s Disease (PD)

The evidence for the involvement of mtDNA variants in PD is quite extensive. For instance, several mtDNA polymorphisms and haplogroups are risk factors for Parkinson’s disease [53,54]. In Caucasian populations, haplogroups J, K and T can reduce the risk of PD, whereas super-haplogroup HV increases the risk of PD [55]. It has also been shown that mitochondrial haplogroup B5 reduces the risk of PD in Taiwanese people of ethnic Chinese background [56]. These studies further establish that changes in mtDNA during human migration may be related to development of PD.

Somatic mtDNA mutations accumulate in the brain with age and could be a central factor in the age-related aspect of neurodegenerative diseases. Although pathogenic mutations that are directly associated with PD have not been found, parkinsonian features have occasionally been observed in mitochondrial disorders. For instance, G11778A (MT-ND4) has been linked with a familial form of PD. This mutation causes a loss in complex I function and is the causative factor for Leber hereditary optic neuropathy (LHON), a disease whose symptoms are usually confined to optic atrophy. This result suggests that hereditary mtDNA mutations can also contribute to adult-onset parkinsonism and neurodegeneration [57]. One does not have to look far to find evidence of this link. In a previous report, mtDNA mutations located in the coding regions of MT-ND2 and MT-ND5 were also found much more often in tissue samples from PD brains than in control brain tissues [58]. The pathogenic variant G13513A, which is the causative factor in LHON/MELAS overlap syndrome, has also been linked to complex I deficiency in Parkinson’s disease [58].

In PD patients, formation of de novo (but low heteroplasmy) MT-ND1, MT-ND2, MT-CO2, MT-CO3, and MT-CYB variants was drastically increased in the SNc relative to the frontal cortex. Thus, in many of these patients, it may be the combined mutational load of many heteroplasmic mutations (mtDNA mutation burden)—accumulated over an entire lifespan—that ultimately triggered PD [59]. Furthermore, the number of identified complex I variants was fewer than expected. This finding suggests that variations in complex I may not be as decisive in the occurrence of PD as was previously thought, or that mutations in complex I components may actually induce apoptosis (thereby selectively removing these cells from the tissue entirely).

The effect of aging on PD risk can be partly explained by the fact that mtDNA deletions are abundant and lead to functional impairments in aged human SNc neurons. High levels of these mtDNA deletions are linked to ETC deficiency, which favors increased oxidative damage, Lewy body formation, and apoptotic cell loss [60,61,62]. Song et al. specifically detected an increased abundance of mtDNA deletions in the SN of mice that had decreased mitochondrial protection by Parkin protein. These deletions produced bioenergetic, neurobehavioral, and dopaminergic deficits that exceeded those of mice with knock-out of *Twinkle* or *Parkin* alone. This suggests that mtDNA deletions may increase selectively with age and, in the process, contribute to dopaminergic neurodegeneration and PD [63]. Oxidative stress-induced mtDNA damage in the SN is also seen in the postmortem brain tissue of PD patients. In fact, mtDNA damage is detectable before degeneration, and such mtDNA damage (particularly to complex I components) may turn out to be a useful molecular marker for vulnerable nigral neurons in PD, [64].

Impairment of mitochondrial function affects ATP synthesis, produces excessive ROS, destroys calcium homeostasis, activates abnormal cell-signaling pathways, and causes serious cellular damage due to the pivotal role of mitochondria in powering the cell. PD-related genes maintain mitochondrial homeostasis; an imbalance in mitochondrial dynamics can cause impaired mitochondrial function and high levels of oxygen free radicals due to dysfunctional mitochondrial autophagy. This condition can produce excessive neurotoxic products, eventually causing PD.

### 3.3. Mitochondrial DNA Variants and HD

The mtDNA is a major site of ROS-induced oxidative damage. In addition to indicating bioenergic dysfunction, damaged mtDNA is associated with neurodegeneration and pathogenesis of the complex respiratory chain activities in HD. Banoei et al. showed that HD patients had higher rates of mtDNA deletions. This finding may indicate that mtDNA deletions are influenced by an extramitochondrial mechanism, and that CAG repeat instability and mutant HTT are causative factors in mtDNA damage [65]. Mutant HTT also appears to cause mitochondrial dysfunction in blood leukocytes. A study by Petersen’s group showed that the mtDNA/nDNA copy number ratio in blood leukocytes increased before and decreased after onset of HD. Thus, fluctuations in this ratio may be a biomarker of HD progression [66].

Additional insights come from an analysis of the D-loop, which accumulates mutations at a higher rate than other mtDNA regions and may prove fertile ground for uncovering mtDNA mutations that can act as additional genetic markers for HD. Sequencing the D-loop regions of HD patients, Mousavizadeh et al. found that C16069T, T16126C, T16189C, T16519C, and C16223T variants tend to be associated with HD, whereas the presence of C16150T, T16086C, and T16195C correlated with a drastically decreased risk of HD [67]. Regarding variants in mtDNA genes, Kasraie et al. screened tRNALeu/Lys and ATPase 6 mutations in 20 Iranian HD patients. One patient showed the A8656G mutation in ATPase 6 (which induces a significant change in protein structure), resulting in defects in ATP production [68].

Mutant HTT affects mitochondrial dynamics and biogenesis in mouse models of HD, further confirming that HD pathogenesis is closely related to these features. OXPHOS dysfunction, mitochondrial fragmentation, and decreased biogenesis are important factors in mutant HTT modification. Mutant HTT may cause disease by affecting mitochondrial dynamics and biogenesis in neurons, and age-related mitochondrial dysfunction may further promote pathological development of HD [24].

### 3.4. Mitochondrial Mutation and Diabetes

Diabetes has been associated with specific mtDNA variants. For instance, the A3243G mutation that causes MELAS was linked with maternally inherited diabetes and deafness (MIDD), which accounts for ~1% of all diabetes cases [69]. In fact, the overtly mitochondrial form of the disorder and the “diabetes” form are different phenotypic manifestations of the same underlying genetic etiology. It should also be noted that all primary mitochondrial diseases have a high risk of comorbidity with diabetes. Thus, it is routine to monitor patients with primary mitochondrial disease for signs of diabetes.

Mutations in mtDNA genes that encode ETC-related components cause disease primarily by reducing OXPHOS efficiency. A3243G mutation occurs within a noncoding gene that produces mitochondrial leucine tRNA. Loss of this tRNA reduces overall protein translation and favors mis-incorporation of amino acids into proteins, leading indirectly to severe ETC deficits. The A3243G mutation [70] causes impaired insulin secretion and insulin resistance [71] by decreasing mitochondrial oxygen consumption and increasing ROS production in the cell, leading to mitochondrial dysfunction. Another well-known variant is the T16189C variant [72] in the D-loop, which is critical for regulation of mitochondrial gene transcription and replication. This variant may affect mitochondrial metabolism, which in turn may hinder ATP production and affect insulin secretion. Therefore, mitochondrial dysfunction is a key pathogenic aspect of diabetes. Mitochondrial abnormalities lead to abnormal energy metabolism, reduced ATP synthesis and abnormal calcium regulation, and ultimately, all of these factors contribute to a reduction in insulin secretion. For these reasons, mitochondria-based diabetes research has become an increasingly important field for those seeking new therapeutic targets for diabetes.

### 3.5. Mitochondrial DNA Variants and Cancer

The overall mutation rate in human mtDNA is estimated to be far higher than the mutation rate in the nuclear DNA. There are several reasons for this discrepancy. First, the ROS-saturated mitochondrial environment creates an elevated risk of DNA damage relative to the less chemically reactive nuclear environment. Second, the mitochondrion-specific DNA polymerase enzyme POLG is much more error-prone than the Family B DNA polymerases that are utilized in the eukaryotic nucleus. Third, mtDNA is replicated at a much greater rate than nDNA and is replicated continually, even in non-dividing cells. Consequently, mutations in mtDNA tend to accumulate and may contribute to various diseases.

Dozens of studies have documented mtDNA mutations that result in OXPHOS inhibition and cancer development, and mtDNA mutations are increasingly being identified in tumors. A recent sequencing study analyzed somatic mtDNA mutations in 1675 tumors, revealing 1907 mtDNA mutations [73]. The common 4977-bp mtDNA deletion has also been observed with a relatively high frequency in blood samples from breast cancer patients, strengthening the link between mtDNA mutation and cancer [74].

The D-loop region is the main region controlling the transcription and replication of mtDNA. Genetic instability of this region may affect regulatory sequences in the noncoding region and ultimately lead to a reduction in mtDNA copy number (mtCN). This decrease is closely linked to a reduction of the OXPHOS complex protein level, resulting in abnormal bioenergetics in tumor cells. Based on examinations of 61 human HCCs and corresponding nontumor liver tissue samples, Lee et al. found that 39.3% of HCCs carried somatic mutation(s) in the mtDNA D-loop, with most of these mutations being homoplasmic [75].

The mtDNA copy number in the cell is precisely controlled by the metabolic output of the ETC. Proteins with a role in mitochondrial transcription, such as transcription factor A, mitochondrial (TFAM), can regulate the mitochondrial replication efficiency. Mutations in these key mtDNA replication proteins can diminish the mtDNA copy number, thereby compromising mitochondrial function by restricting its production of the RNA and protein molecules necessary for normal function. Researchers have shown that a truncation mutation of *TFAM* mediates apoptosis by downregulating cytochrome b transcription and, presumably, by decreasing the interaction with the mitochondrial heavy-strand promoter. These findings support the role of TFAM and mitochondrial stability in colorectal cancer tumorigenesis [76]. Since tumor cells typically have a lower mtDNA copy number than normal tissues, this reduced mtDNA copy number is linked to key "driver" mutations that lead to cell carcinogenesis [77].

## 4. The Relationship between mtDNA Mutations and Aging

### 4.1. Accumulation of mtDNA Mutations in Normal Aging

We have previously shown that the rate of somatic mtDNA defects increases significantly with age. Many such mutations are nonsynonymous or located in RNA-encoding gene regions, leading to respiratory chain dysfunctions. Moreover, induced pluripotent stem cells (iPSCs) from older individuals may carry large amounts of mtDNA mutations [78]. We isolated ~1 million mixed fibroblasts from a skin biopsy of a 72-year-old individual and performed mtDNA sequencing using the Illumina MiSeq sequencing platform. Skin fibroblasts were subcloned, and 10 cloned fibroblasts were randomly selected for analysis. We detected 34 mutations, 14 of which were at a heteroplasmy level of >15%, with 6 mutations exceeding 80%. These results strongly suggest that most mtDNA defects are acquired from in vivo sources, and that the number of mtDNA mutations increases sharply with age, especially for individuals over 60 years. In other words, an increase in mtDNA mutation appears to be occurring that broadly correlates with the greater damage to cell function that comes with age [78]. However, since this data is derived from clonal cell lines propagated extensively in cell culture, it remains unclear how closely these results match the heteroplasmy levels that these mutations achieve in vivo. That such mutations are acquired with age is clear, but whether or not they reach meaningful levels of heteroplasmy under natural conditions of cellular proliferation and turnover remains an open question.

Oxidative damage from reactive oxygen species leads to DNA damage as the result of modifications of purine and pyrimidine bases, cleavage of single- or double-stranded DNA, and cross-linkage with other molecules. Although damage to nDNA generally receives more attention, damage can also occur quite readily in the mtDNA, given its proximity to the main source of ROS production in the cell. There is a well-established correlation between aging and a decline in mitochondrial function, which likely contributes to age-related senescence and geriatric disease. *Polg^D257A^* “mutator” mice—which exhibit increased mtDNA mutation rates—show an accelerated aging phenotype, suggesting that the accumulation of mtDNA mutations over time may be a crucial factor driving the aging of mammals [79].

The age-related decline in mitochondrial function is likely caused, in large part, by the gradual accumulation of somatic mtDNA mutations due to ROS damage and DNA replication errors. This accumulation of mtDNA mutations promotes even more ROS production, establishing a vicious cycle and accelerating the aging process. Although a serious mtDNA mutation can be acquired early in life, for most people it will take several decades to acquire one or more disease-causing mutations and have them reach a sufficiently high level of heteroplasmy to cause serious health issues. This gradual accumulation of mutations and increase in heteroplasmy may help explain the time-dependent decline in function that occurs with age. Niemann et al. reported that a cytochrome c oxidase mutation encoded in the mtDNA accelerated the senescence of liver organs in mice by interfering with the ROS response [80]. DeBalsi et al. have summarized the theory that oxidative damage promotes the aging phenotype, suggesting that functional defects in nuclear-encoded mtDNA replication proteins—including Pol γ, PolG2, Twinkle, TFAM, MGME1, and RNase H1—lead to aging-related diseases [81].

Somatically-acquired mtDNA mutations can occur anywhere in mtDNA and may even include large deletions or duplications. Loss of mtDNA integrity (by altered mtDNA copy number or increased mutations) has been implicated in cellular dysfunction with aging [82]. Deletion mutations are an insidious risk because the reduced size of mtDNA molecules carrying large deletions (ΔmtDNAs) gives them a replicative advantage over normal mtDNA. Neuhaus et al. demonstrated that as mice or humans age, their adrenal medullas accumulate large numbers of mtDNA deletions, leading to mitochondrial dysfunction accompanied by apoptosis, severe inflammation, and marked fibrosis [83]. In a study of Sod1 (−/−) mutant mice, researchers found that mitochondrial deletion mutations differ from tissue to tissue, with some tissues possessing intrinsically higher metabolic levels showing the increased presence of mtDNA deletion mutations and DNA oxidative damage [84]. These findings have also been observed in sequencing analyses of skeletal and cardiac muscle tissue cells. This progressive expansion in ΔmtDNAs may help to explain the observed aging-related loss in muscle tone and strength [85]. Foote et al. have found that mitochondrial respiration in arterial tissues decreases with age, and that reduced mitochondrial function and mtDNA integrity promotes vascular aging. Furthermore, they showed that rescuing mtDNA integrity through overexpression of the mitochondrial helicase Twinkle increased mtDNA copy number and improved mitochondrial function in mouse arteries [86]. In a separate study measuring mtDNA copy number in blood cells from 1067 subjects aged 18 to 93 years, Mengel-From et al. found that a lower mtDNA copy number appeared to accompany advanced age [87].

### 4.2. Factors Contributing to the Somatic mtDNA Mutation Rate

Given the body of data correlating aging and mtDNA mutations, just how extensive are somatic mtDNA mutations? This process is likely to be fundamentally different from age-related nDNA damage, as mitochondrial genetics involves unique considerations that are distinct from the Mendelian genetics of the nuclear genome.

First, we must consider the number of copies of each genetic element in the body. Several estimates have been made for the total number of cells in a “standard” human adult, ranging anywhere from ~3.0 × 10^13^ [88,89] to ~3.72 × 10^13^ cells per individual [90]. By comparing the nDNA and mtDNA copy numbers in various tissues, the average cell is thought to possesses anywhere from 100 to 10,000 copies of mtDNA per cell depending on the tissue or cell type [91], with most cells being in the middle to upper end of this range. In other words, the average human adult carries anywhere from 10^16^ to 10^17^ copies of mtDNA. Furthermore, in the context of cell division, Bogenhagen and Clayton long ago observed that, “The number of mitochondrial DNA molecules in a cell population doubles at the same rate as the cell generation time” [92]. This means that, except for very limited and specific circumstances, both mtDNA and nDNA must double their number with each round of cell division, so that each daughter cell receives the appropriate complement of DNA. Thus, for the purposes of our calculation, every time a cell population doubles, the mtDNA population must double as well.

Accounting for the number of cell divisions required to turn a zygote into fully developed newborn, we can estimate the average number of times that each mtDNA molecule replicates during development. Using the 3.72 × 10^13^ estimate from above, we postulate that it takes at least 45 rounds of cell division to produce a child from the single-celled zygote, which implies that the average mtDNA molecule has undergone at least 45 rounds of replication to produce the corresponding amount of mtDNA. This does not account for cell loss due to apoptosis during development, which can account for upwards of 50% of cells in the nervous system [93]. However, apoptosis is not likely to be a decisive factor in the overall calculations (at least for development), as even compensating for a loss of 50% of the final number of cells would only require an adjustment of one additional round of replication. Also, apoptosis affects nDNA and mtDNA equally and, therefore, is unlikely to lead to major differences between the two genetic elements. Thus, for the purpose of simplification, a conservative estimate of 45 rounds of replication per cell division will be assumed to be roughly accurate.

Next, we estimate the accumulation of somatic mtDNA mutations, starting with the fate of preexisting maternally-inherited mtDNA heteroplasmies. Although most such heteroplasmies will be present at very low frequencies, here we are more concerned with the potential for radical shifts in the percentage of heteroplasmy as cells divide over time. Unlike nuclear alleles that exist in a relatively static ratio after fertilization (barring chromosomal rearrangement or de novo mutations), mtDNA variants routinely undergo random and directed shifts in frequency, both within individuals and across generations. In the germline, a major factor in heteroplasmic shifts between generations is the “mtDNA bottleneck”, wherein the mtDNA copy number undergoes a drastic reduction followed by a compensatory increase [21]. These mtDNA bottleneck events can create massive distortions relative to the initial maternal heteroplasmy levels.

Even given the low initial percentage of heteroplasmy in the zygote, 45 rounds of cell division can result in substantial increases in this percentage, particularly if there is a proliferative advantage favoring the mutant variant. For example, we previously estimated that a variant present at 5% heteroplasmy in the zygote and possessing a 20% proliferative advantage over the wild-type mtDNA variant would be expected to reach >70% heteroplasmy after 45 rounds of cell division [94]. This estimate is well within the range for disease presentation for most known disease-causing mtDNA variants. A 40% advantage would be projected to lead to near-homoplasmy for the variant, demonstrating the potentially decisive role that proliferative differences likely play in determining the final state of somatic mtDNA.

Mitochondrial turnover is constant, even in post-mitotic cells such as neurons, with old or damaged mitochondria continually removed through mitophagy. In the process, mtDNA molecules carried by these mitochondria are destroyed along with their organellar host. In fact, mtDNA molecules are continually lost even in intact mitochondria, largely due to damage by ROS and other chemical processes. The actual rate of mtDNA turnover is the subject of some controversy, with the estimated half-life anywhere from days to as long as a year [95]. The mtDNA half-life is likely to be affected by the energy requirements of the specific cell type; more intensive ATP production will inevitably produce higher levels of ROS, leading to a higher rate of mtDNA damage and turnover. Considering the most conservative estimate of 12 months for the mtDNA half-life, the average non-proliferative cell will need to replicate its entire mtDNA complement ~70 times (i.e., undergo 70 population doublings) by 70 years just to maintain a steady mtDNA copy number. At the other extreme, if one considers one of the shorter published estimates of 12 days [96], ~2131 rounds of replication will occur by the age of 70 years. In truth, the figure is likely to differ substantially from tissue to tissue. For our purposes, we will assume a conservative half-life estimate of 3 months, in line with the predicted half-life “in the order of months” suggested by Poovathingal et al. [95]. This would imply ~280 rounds of replication between birth and the age of 70 years.

Next, we will estimate the rate of acquisition for somatic mtDNA mutations. First, consider a somatic mtDNA mutation acquired in a neural precursor cell early in development and present at an average frequency of 1% in the post-mitotic neural population. Although many mtDNA variants would be expected to undergo random genetic drift, we will assume that this mtDNA variant will lead to differences in proliferative potential. This is not an unreasonable assumption for a theoretical calculation, as polymorphic sequences in the noncoding D-loop region of mtDNA can cause significant differences in mtDNA replication efficiency in embryo-derived stem cell lines [97]. Given these assumptions, if the mtDNA variant confers even a tiny proliferative advantage (for example, a 2% increase per round of mtDNA doubling), most neurons would contain ~14% heteroplasmy of the mutant mtDNA allele by the age of 70 years. If the starting frequency in this scenario were 10%, then the mutation would attain >60% heteroplasmy by 70 years of age. On the other hand, increasing the proliferative advantage to approximately 5% per round of replication would mean that the mutant variant would be at >90% heteroplasmy by the age of 70, even with a starting frequency of 1%.

Age-related mtDNA mutations are most commonly observed and studied in post-mitotic tissues such as muscles or neurons [98], suggesting post-mitotic tissues are more susceptible to the accumulation of mtDNA mutations than proliferating cell types. There are a variety of reasons why this might be the case; for instance, cells carrying mtDNA mutations may simply be outcompeted by their healthier counterparts in proliferating cell types. However, there have been a few studies showing that such mutations can also accumulate faster with age in human colonic crypt cells [99] and in mouse hematopoietic stem cells from the C57BL/6 background [17], both of which represent highly proliferative cell types. Thus, it is worth considering the impact that cellular proliferation may have on the accumulation of mtDNA mutations with age.

In highly proliferative tissues, the rate of production of new cells can be as high as 500 billion cells per day (e.g., in bone marrow), to compensate for cells lost to apoptosis, infection, physical destruction, etc. [100]. Mathematical modeling has indicated that producing this many mature cells from a population of ~80 million hematopoietic stem cells (HSCs) requires 16–19 rounds of replication, depending on the cell lineage [101]. To maintain a population of 80 million HSCs while sustaining the production of peripheral blood cells, HSCs must replicate themselves once every 40 weeks [102] (or ~91 times by the age of 70). This estimate adds an additional 91 rounds of mtDNA replication to our baseline estimate of 280 rounds that occur in nonproliferating tissues, for a total of 371 rounds of mtDNA replication by the age of 70. Assuming a 1% starting frequency and 2% proliferative advantage for an mtDNA variant, this math would translate to a nearly 28% mutant variant frequency by the age of 70 (vs. 14% in nonproliferating cells). This exercise clearly demonstrates the elevated heteroplasmy risk present in proliferating vs. nonproliferating tissues.

This analysis assumes that no new mtDNA variants are introduced over time, an impossible assumption given the elevated mutation rate of mtDNA. Much of this increased error rate can be attributed to intrinsic errors of the mitochondrial replication machinery, although some mutations are probably induced by ROS-mediated damage. Although a direct calculation of the somatic mtDNA mutation rate per round of replication is difficult to precisely establish, a reasonable estimate may be inferred from *in vitro* experiments on the mitochondrial DNA polymerase, POLG. Based on a thorough analysis of the fidelity of the POLG enzyme and its proofreading exonuclease, the error rate has been estimated to be approximately 1 misincorporated base per 1.8 × 10^6^ to 3.6 × 10^7^ bases synthesized [103]. If an average cell contains 1000 copies of mtDNA, with each molecule 16,569 base pairs long [104,105], this math would roughly translate to an average of 0.5 to 9.2 new mtDNA mutations per round of cell division. Thus, given the minimum estimate of 45 rounds of cell division that must occur (on average) to produce the typical human adult, an average adult cell would be expected to begin with anywhere from 22 to 414 de novo mutations in its mtDNA population, suggesting that a significant portion of the mtDNA population could possess at least one de novo mtDNA mutation. Although most of these mutations will be present at frequencies far below the threshold to cause any disease phenotypes, additional mutations will continue to accumulate with age, particularly in proliferating cell populations.

Any mtDNA molecules containing a deleterious mutation in a coding gene, combined with a D-loop mutation that creates a strong proliferative advantage, would have the potential to create severe issues in old age, particularly if they cause the cell to take on neoplastic properties. Furthermore, this ubiquitous and steadily accumulating mutational load in the mtDNA population will likely create general problems for the ETC function and mitochondrial health, thus contributing to age-related senescence. For these reasons, we believe there is great merit in the argument that mitochondrial defects play a significant role in the development of many common diseases of age. We hypothesize that the diversity of mutations in mtDNA could be decisive for the variability of clinical phenotypes, such as age of disease onset.

### 4.3. Calculating the mtDNA Mutation Rate with Age

We summarize our assumptions here:

**(1) POLG Error rate**: The combined fidelity of POLG and its proofreading exonuclease leads to an estimated error rate between 1 in 1.8 × 10^6^ bp and 1 in 3.6 × 10^7^ bp [103]. This translates to a per nucleotide error rate falling in the interval [2.8 × 10^−8^; 5.6 × 10^−7^], which we will refer to as “*µ*”. We will also need to account for the fact that each daughter cell in a cell division has a 50% chance of receiving any particular de novo mutation, as well as the fact that any newly generated mutation has a 50% chance of being lost during an mtDNA half-life. Thus, the mutation rate will need to be divided by two to represent the effective mutation rate.

**(2) mtDNA copy number per cell**: The number of mtDNA copies per cell, “*Z*”, is between 100 and 10,000, which varies substantially between different cell types. For this example, we suggest *Z* = 1000 as a conservative estimate, which is the minimum number of copies observed in most cell types [106].

**(3) Number of bases in a single mtDNA molecule**: The number of base pairs in a single human mtDNA molecule is assumed to be the reference genome value of 16,569 bp [104,105].

**(4) Number of replication events between fertilization and birth**: Let *X*_1_ = the number of mtDNA replications between fertilization and birth. As described in the previous section, we consider 45 to be the minimum estimate for this value across all cell types (although there will be considerably variation based on the particular cell type in question).

**(5) Number of replication events between birth and a given age in non-proliferating cells:** Let *X*_2_ = the number of mtDNA replications between birth and a given age (in this case, age 70) in a non-dividing cell. This number will be almost exclusively driven by the replacement of mtDNA lost to mtDNA turnover. Based on a conservative estimate of an mtDNA half-life of 3 months [95], we will set this value to 280 at the age of 70.

**(6) Number of additional replications in proliferating cells: ***X*_3_ = the additional number of mtDNA replications in proliferating cells between birth and a given age. Of course, for non-proliferating cells, this *X*_3_ = 0. In this case, we will assume that the individual in question is 70 years old, and that the cells in question are HSCs that are replicating once every 40 weeks, on average [102]. The total number of replications for proliferating cells *Y* = *X*_1_ + *X*_2_ + *X*_3_. From the existing evidence we outlined in the previous section, we can induct that approximately *E*(*X*_1_) = 45, *E*(*X*_2_) = 280, and for proliferating cells (such as HSCs), *E*(*X*_3_) = 91. Therefore, *E*(*Y*) = 416.

Based on these four parameters, an estimate for the mtDNA mutation rate at a given age can be calculated. We can estimate that in an average proliferating cell, the number of de novo mutations, *B* = 16,569ZY*µ*/2. If *µ*, *Z*, and *X_i_*’s are all independent, then *E*(*B*) = 16,569 ⋅*E*(*Z*) ⋅ *E*(*µ*/2) ⋅ *E*(*Y*).

Now we model each of the above quantities as a random variable. As a first approximation, we postulate that the POLG error rate *µ* ~ Unif(2.8 × 10^−8^, 5.6 × 10^−7^). To impose a distribution on *Z*, we will set *Z* = 1000 (as described above). So we assume a uniform distribution over the interval [2,4] for *logZ* (log-base 10). Therefore, the expected value of *logZ* is 3, which transforms back to *Z* = 1000. For now, we assume the number of bp in a single mtDNA molecule is a constant. Let *X*_1_ ∼ Unif(10, 80), with a mean 45; *X*_2_ ∼ Unif(180, 380), with a mean 280; and *X*_3_ ∼ Unif(81, 101), with a mean 91. For a different given age, *X*_2_ and *X*_3_ will have different distributions. Thus, the total number of mutations per proliferating cell as age 70 is,
*B* = 16,569*Z* (*X*_1_ + *X*_2_ + *X*_3_) *µ*/2

Under this setting, assuming all variables are independent of each other, the expected mutation burden is,
*E*(*B*) = 16,569 ⋅ 10*^E^*^(l*ogZ*)^*E*(*µ*/2) (*E*(*X*_1_) + *E*(*X*_2_) + *E*(*X*_3_))     = 16,569 × 1000 × 1.47 × 10^−7^ × (45 + 280 + 91) = 1013

Note that we take the geometric mean of *Z*. We can also calculate mean mutation rate per cell by fixing some parameters. For example, for the low and high estimate for the POLG error rates [103],
*E*(*B*|*µ* = 2.8 × 10^−8^) = 97
*E*(*B*|*µ* = 5.6 × 10^−7^) = 1930

### 4.4. Simulating mtDNA Mutations

Based on the framework described in the previous section, we can use simulation to characterize the distribution of quantity *B*, the per (proliferating) cell mutation burden at the age of 70. We can, of course, use any set of distributional assumptions. Figure 1 is the frequency histogram of quantity *B* under the current distributional setting. Figure 2 is the 1 minus cumulative distribution function of *B*, which indicates the probability of at least *x* mutations per cell. Both proliferating and non-proliferating curves are plotted. The actual probabilities for a selected number of mutations per cell are tabulated in Table 1.

It is interesting to note that under the assumptions that we have described here, almost 90% of non-proliferating cells would be expected to have at least 100 mutations per cell by the age of 70, and almost no cells would have fewer than 10 mutations (Table 1). This would seem to represent a non-trivial mtDNA mutational burden, and would seem to bolster the hypothesis that mtDNA mutations are a significant contributor to the aging process. However, it is important to note that others in the field may have valid criticisms of the assumptions that we have laid out here, and future data may provide further refinements on the estimations of cellular proliferation rate, mtDNA half-life, POLG fidelity, etc. For this reason, the equation that we have described here has been designed to be quite flexible, with variables that can be adjusted to account for a wide range of different assumptions or scenarios. It is our hope that the framework provided by this equation will serve as a starting point for others in the field to provide their own input on what the true mtDNA mutational burden might be.

## 5. Conclusions

In this review, we have shown that accumulating evidence strongly supports the hypothesis that aging is linked to mitochondrial damage, mainly due to the progressive accumulation of mutant mitochondria. Most individuals will acquire at least some deleterious mtDNA mutations over their lifetime, and many of the diseases that we associate with advanced age may be a direct consequence of this accumulated mutational load. This notion is further supported by the relatively high mtDNA mutation rate, which makes the acquisition of de novo mtDNA mutations over time a virtual certainty. Even if each of these individual mutations is present at a low frequency, the combined mutational burden that they represent in aggregate may itself lead to cellular dysfunction and disease. It is also worth considering the possibility that certain variants can contribute to the appearance of more common “non-mitochondrial” diseases (e.g., cancer and diabetes) at frequencies below the traditional threshold for primary mitochondrial disease. For these reasons, accurate and sensitive mitochondrial sequencing techniques that can detect disease-causing variants at very low heteroplasmy are likely to become an important part of the physician’s arsenal. Such tools will allow more informed treatment decisions to be reached at an earlier point in the disease progression, giving patients the maximum possible benefit.

## Figures and Tables

**Figure 1 cells-08-00608-f001:**
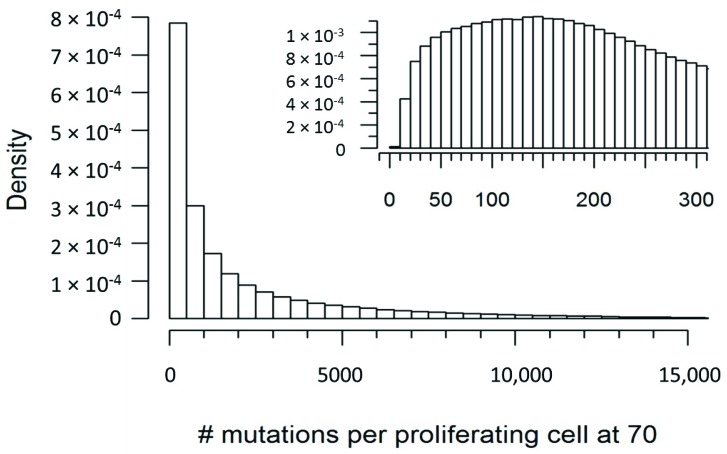
Simulated distribution of mutations per proliferating cell at the age of 70.

**Figure 2 cells-08-00608-f002:**
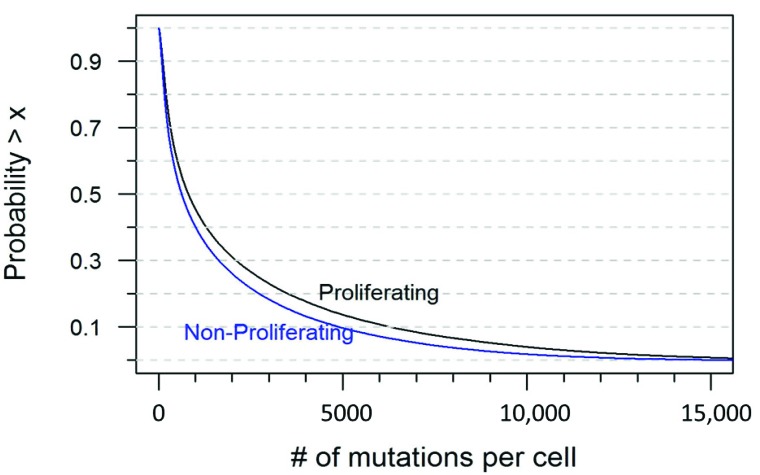
Reversed cumulative distribution of mutations per cell at the age of 70.

**Table 1 cells-08-00608-t001:** Probability of a given number of mutations per cell.

Number of Mutations	Probability in a Proliferating Cell	Probability in a Non-Proliferating Cell
≤10	<0.001	0.001
≤50	0.031	0.050
≤100	0.084	0.118
≤200	0.195	0.250
≥500	0.606	0.550
≥1000	0.455	0.402
≥1500	0.370	0.318
≥2000	0.311	0.261
≥3000	0.231	0.183
≥5000	0.137	0.097

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
