# Peer review of "Mitochondrial DNA Variants and Common Diseases: A Mathematical Model for the Diversity of Age-Related mtDNA Mutations"

_cells, 2019, doi:10.3390/cells8060608_

Round 1
Reviewer 1 Report
Li et al.“Mitochondrial Dysfunction and Common Diseases – A Math Model for the Diversity of Age-related mtDNA Mutations. “
Li et al.review how mitochondrial dysfunction and mtDNA mutations contribute to a variety of different neurodegenerative diseases, cancer, diabetes, and aging. They next go on to provide rationale and mathematical modeling to estimate the number of mtDNA mutations that may accumulate over a person’s lifespan. The authors account for rates of cellular proliferation, polymerase gamma error rates, mtDNA replication rates, mitophagy and mitochondrial turnover as factors in their model.
Major Concerns:
1. A majority of post-mitotic age related mtDNA mutations and deletions occur in post-mitotic tissues such as skeletal muscle fibers, neurons, and cardiomyocytes (Larsson, 2010). The only exception to this is colonic crypts, but these mutations proliferate after differentiation of the colonic stem cell (Taylor et al., 2003). Colonic stem cells do not turn over and the colonic stem cell accumulates damage over time with age. It is unclear how this mathematical model which heavily relies on cellular proliferation (which the authors state their estimation of mtDNA mutation frequency would be higher in proliferating tissues (pg.30) is relevant to the actual mtDNA mutation load found tissues in vivo.
2. The authors cite a study that they were apart of when they found high levels of heteroplasmy in adult IPSCs from aged individuals as rational for their mathematical model. The skin biopsy and blood of elderly patients where an individual cell was used to generate a subclone to convert into an IPSC showed slightly elevated levels of heteroplasmy (Kang et al., 2016). However, these mutations really propagated more so in culture and after stem cell conversion. It is clear that mtDNA mutations accumulate in dermal fibroblasts with aging in the organism; however, it is unclear when accounting for cellular turnover and replacement if they reach significantly high levels of heteroplasmy outside of the laboratory setting.
Concerns:
1. The grammar and wording needs to be rechecked and corrected for this manuscript.
2. There are large areas of text that are lacking in-text citations.
3. The first part of the review that introduces the different neurodegenerative diseases, cancer, diabetes, and aging should focus solely on the presence of mtDNA mutations in these varying diseases instead of going in-depth on how different types of mitochondrial dysfunction have been associated with those diseases. It is distracting to the focus of the review.
Kang, E., Wang, X., Tippner-Hedges, R., Ma, H., Folmes, C.D.L., Gutierrez, N.M., Lee, Y., Van Dyken, C., Ahmed, R., Li, Y., Koski, A., Hayama, T., Luo, S., Harding, C.O., Amato, P., Jensen, J., Battaglia, D., Lee, D., Wu, D., Terzic, A., Wolf, D.P., Huang, T., Mitalipov, S., 2016. Age-related accumulation of somatic mitochondrial DNA mutations in adult-derived human ipscs. Cell Stem Cell. https://doi.org/10.1016/j.stem.2016.02.005
Larsson, N.-G., 2010. Somatic Mitochondrial DNA Mutations in Mammalian Aging. Annu. Rev. Biochem. https://doi.org/10.1146/annurev-biochem-060408-093701
Taylor, R.W., Barron, M.J., Borthwick, G.M., Gospel, A., Chinnery, P.F., Samuels, D.C., Taylor, G.A., Plusa, S.M., Needham, S.J., Greaves, L.C., Kirkwood, T.B.L., Turnbull, D.M., 2003. Mitochondrial DNA mutations in human colonic crypt stem cells. J. Clin. Invest. https://doi.org/10.1172/JCI19435
Author Response
Response to Reviewer 1Li et al. “Mitochondrial Dysfunction and Common Diseases – A Math Model for the Diversity of Age-related mtDNA Mutations.”
Li et al.review how mitochondrial dysfunction and mtDNA mutations contribute to a variety of different neurodegenerative diseases, cancer, diabetes, and aging. They next go on to provide rationale and mathematical modeling to estimate the number of mtDNA mutations that may accumulate over a person’s lifespan. The authors account for rates of cellular proliferation, polymerase gamma error rates, mtDNA replication rates, mitophagy and mitochondrial turnover as factors in their model.
Response: Thank you for your careful reading of our review, and for the insightful comments provided below. We hope that the current revision has adequately addressed those concerns.
Major Concerns:
1. A majority of post-mitotic age related mtDNA mutations and deletions occur in post-mitotic tissues such as skeletal muscle fibers, neurons, and cardiomyocytes (Larsson, 2010). The only exception to this is colonic crypts, but these mutations proliferate after differentiation of the colonic stem cell (Taylor et al., 2003). Colonic stem cells do not turn over and the colonic stem cell accumulates damage over time with age. It is unclear how this mathematical model which heavily relies on cellular proliferation (which the authors state their estimation of mtDNA mutation frequency would be higher in proliferating tissues (pg.30) is relevant to the actual mtDNA mutation load found tissues in vivo.
Response: Thank you for providing this additional nuance to the discussion of age-related mtDNA accumulation. You are correct in your assessment that many of the tissues relevant to mtDNA mutation are post-mitotic tissues; however, our intention in this manuscript is to create a mathematical framework for evaluating mtDNA mutation accumulation in all possible types of tissues. It is also not clear that proliferating tissues are entirely free from the risk of accumulating mtDNA mutation; for instance, Yao et al. (2007, PMID: 17185390) have demonstrated a clear, age-related accumulation of mtDNA mutations in mouse hematopoietic stem cells from the C57BL/6 background. Thus, we would like our mathematical framework to be able to account for any possible rate of proliferation so it can be applied to all possible tissues and biological situations.
2. The authors cite a study that they were apart of when they found high levels of heteroplasmy in adult IPSCs from aged individuals as rational for their mathematical model. The skin biopsy and blood of elderly patients where an individual cell was used to generate a subclone to convert into an IPSC showed slightly elevated levels of heteroplasmy (Kang et al., 2016). However, these mutations really propagated more so in culture and after stem cell conversion. It is clear that mtDNA mutations accumulate in dermal fibroblasts with aging in the organism; however, it is unclear when accounting for cellular turnover and replacement if they reach significantly high levels of heteroplasmy outside of the laboratory setting.
Response: This is a fair point; it is indeed unclear just how much the laboratory data reflects the reality in vivo. Thus, we have revised the text to make this clear to the reader.
Concerns:
1. The grammar and wording needs to be rechecked and corrected for this manuscript.
Response: The grammar has been checked and revised as needed in the new version of the manuscript.
2. There are large areas of text that are lacking in-text citations.
Response: We hope that the in-text citation gaps have been addressed by the new citations added to the revised manuscript. The splitting of the manuscript into two smaller reviews may also help somewhat with this issue by providing more focus to the text.
3. The first part of the review that introduces the different neurodegenerative diseases, cancer, diabetes, and aging should focus solely on the presence of mtDNA mutations in these varying diseases instead of going in-depth on how different types of mitochondrial dysfunction have been associated with those diseases. It is distracting to the focus of the review.
Response: This is a good point. We have tried to resolve this issue by splitting the manuscript into two smaller reviews: one focused on the nuclear genes involved in mitochondrial disease and aging, and one focused entirely on the mtDNA mutations and aging.
Kang, E., Wang, X., Tippner-Hedges, R., Ma, H., Folmes, C.D.L., Gutierrez, N.M., Lee, Y., Van Dyken, C., Ahmed, R., Li, Y., Koski, A., Hayama, T., Luo, S., Harding, C.O., Amato, P., Jensen, J., Battaglia, D., Lee, D., Wu, D., Terzic, A., Wolf, D.P., Huang, T., Mitalipov, S., 2016. Age-related accumulation of somatic mitochondrial DNA mutations in adult-derived human ipscs. Cell Stem Cell. https://doi.org/10.1016/j.stem.2016.02.005
Larsson, N.-G., 2010. Somatic Mitochondrial DNA Mutations in Mammalian Aging. Annu. Rev. Biochem. https://doi.org/10.1146/annurev-biochem-060408-093701
Taylor, R.W., Barron, M.J., Borthwick, G.M., Gospel, A., Chinnery, P.F., Samuels, D.C., Taylor, G.A., Plusa, S.M., Needham, S.J., Greaves, L.C., Kirkwood, T.B.L., Turnbull, D.M., 2003. Mitochondrial DNA mutations in human colonic crypt stem cells. J. Clin. Invest. https://doi.org/10.1172/JCI19435

Reviewer 2 Report
In this review entitled “Mitochondrial dysfunction and common diseases - a math model for the diversity of age-related mtDNA mutations” the Authors discuss the role of mitochondrial defects in aging, cancer, diabetes, Parkinson, Alzheimer and Huntington disease and propose a mathematical formula to calculate the accumulation of mtDNA mutations with age. This review is well written, nevertheless, I have comments in order to improve the manuscript.
Introduction
I suggest adding some sentences to introduce mitochondrial haplogroups, mitochondrial dynamics and mitophagy.
Line 37 Reference [1] should be replaced by “Gray MW, Burger G, Lang BF. Mitochondrial evolution. Science. 1999;283(5407):1476–1481”.
Line 67 I suggest adding a reference.
2. Neurodegenerative diseases
Line 91 I suggest adding a reference.
Paragraph 2.1.3
The relation between mtDNA mutations and AD needs to be discussed more extensively. I suggest these papers: Mitochondrial DNA mutations increase in early stage Alzheimer disease and are inconsistent with oxidative damage. Hoekstra JG et al. Ann Neurol. 2016 80: 301-6; Soltys DT et al. Lower mitochondrial DNA content but not increased mutagenesis associates with decreased base excision repair activity in brains of AD subjects. Neurobiol Aging. 2019 73:161-170.
Line 290 reference 55 does not match the text.
Line 310 I suggest adding a reference
4. Cancer
Line 467 I suggest adding a reference
5. Aging
Line 570 I suggest adding a reference
Line 599 reference 114 does not match the text.
Line 654 Neuhaus et al. The reference is missing
Line 704 I suggest adding a reference
Author Response
Response to Reviewer 2In this review entitled “Mitochondrial dysfunction and common diseases - a math model for the diversity of age-related mtDNA mutations” the Authors discuss the role of mitochondrial defects in aging, cancer, diabetes, Parkinson, Alzheimer and Huntington disease and propose a mathematical formula to calculate the accumulation of mtDNA mutations with age. This review is well written, nevertheless, I have comments in order to improve the manuscript.
Response: We appreciate that our review was well received, and have worked to address your remaining concerns in the revised manuscript.
Introduction
I suggest adding some sentences to introduce mitochondrial haplogroups, mitochondrial dynamics and mitophagy.
Response: We agree that these concepts should be further explained and have added them to the Introductions.
Line 37 Reference [1] should be replaced by “Gray MW, Burger G, Lang BF. Mitochondrial evolution. Science. 1999;283(5407):1476–1481”.
Response: Thank you for the suggestion. The reference has been replaced.
Line 67 I suggest adding a reference.
Response: An appropriate reference has been added.
2. Neurodegenerative diseases
Line 91 I suggest adding a reference.
Response: An appropriate reference has been added.
Paragraph 2.1.3The relation between mtDNA mutations and AD needs to be discussed more extensively. I suggest these papers: Mitochondrial DNA mutations increase in early stage Alzheimer disease and are inconsistent with oxidative damage. Hoekstra JG et al. Ann Neurol. 2016 80: 301-6; Soltys DT et al. Lower mitochondrial DNA content but not increased mutagenesis associates with decreased base excision repair activity in brains of AD subjects. Neurobiol Aging. 2019 73:161-170.
Response: Thank you for the suggestion. We have added this to the revised manuscript.
Line 290 reference 55 does not match the text.
Response: Reference 55 has been replaced with more relevant citations.
Line 310 I suggest adding a reference
Response: Appropriate references have been added.
4. Cancer
Line 467 I suggest adding a reference
Response: An appropriate reference has been added.
5. Aging
Line 570 I suggest adding a reference
Response: An appropriate reference has been added.
Line 599 reference 114 does not match the text.
Response: This reference has been replaced with a more relevant citation.
Line 654 Neuhaus et al. The reference is missing
Response: Thank you for bringing this to our attention. The reference has now been fixed.
Line 704 I suggest adding a referenceResponse: An appropriate reference has been added.
Round 2
Reviewer 1 Report
Sloane et al.“Mitochondrial DNA Variants and Common Diseases – A Mathematical Model for the Diversity of Age-related mtDNA Mutations. “
Sloane et al.review how mtDNA mutations contribute to a variety of different neurodegenerative diseases, cancer, diabetes, and aging. This revision is much more focused that the previous submission. They provide rationale and mathematical modeling to estimate the number of mtDNA mutations that may accumulate over a person’s lifespan. However, it is unclear what the physiological relevance is to this calculation because it doesn’t reflect mtDNA mutations found in biology (mitotic v. post-mitotic), and the mtDNA replication rate is not a good variable for determining the mtDNA mutational rate in cells (Battersby and Shoubridge, 2001; Jenuth et al., 1997; Jokinen et al., 2010). Also, cell cycle and mtDNA replication are not linked (Bogenhagen and Clayton, 1977).
Major Concerns:
1. After the authors have split the original manuscript into two reviews, now this review is lacking a basic introduction to the diseases that it focuses on for the non-expert. It assumes the reader is familiar with all of the neurodegenerative and age-related diseases discussed.
2. In the paragraph starting at lines 350, it unclear why the authors think that the mutant will have a proliferative adaptive during development. If anything, mutant mtDNA (especially mutations in protein coding genes) does not have a proliferative advantage (Fan et al., 2008; Hill et al., 2014).
Concerns:
1. The grammar and wording again still needs to be rechecked and corrected for this manuscript.
2. Lines 117-118 is not correct. Dementia is a description of symptoms that can be reversible (drug use, vitamin or thyroid deficiency, etc.) or irreversible (AD, FTD, etc.). It is not a potential precursor to AD.
3. Lines 186-191 are also not correct. There is not an elevated risk of HD. This is a genetic autosomal dominant disease with either complete or incomplete dominance.
4. Line 211, define IR. The reviewer thinks you are referring to insulin resistance.
Battersby, B.J., Shoubridge, E. a, 2001. Selection of a mtDNA sequence variant in hepatocytes of heteroplasmic mice is not due to differences in respiratory chain function or efficiency of replication. Hum. Mol. Genet. https://doi.org/10.1093/hmg/10.22.2469
Bogenhagen, D., Clayton, D.A., 1977. Mouse L cell mitochondrial DNA molecules are selected randomly for replication throughout the cell cycle. Cell. https://doi.org/10.1016/0092-8674(77)90286-0
Fan, W., Waymire, K.G., Narula, N., Li, P., Rocher, C., Coskun, P.E., Vannan, M.A., Narula, J., Macgregor, G.R., Wallace, D.C., 2008. A mouse model of mitochondrial disease reveals germline selection against severe mtDNA mutations. Science (80-. ). 319, 958–962. https://doi.org/10.1126/science.1147786
Hill, J.H., Chen, Z., Xu, H., 2014. Selective propagation of functional mitochondrial DNA during oogenesis restricts the transmission of a deleterious mitochondrial variant. Nat. Genet. https://doi.org/10.1038/ng.2920
Jenuth, J.P., Peterson, A.C., Shoubridge, E.A., 1997. Tissue-specific selection for different mtDNA genotypes in heteroplasmic mice. Nat. Genet. https://doi.org/10.1038/ng0597-93
Jokinen, R., Marttinen, P., Sandell, H.K., Manninen, T., Teerenhovi, H., Wai, T., Teoli, D., Loredo-Osti, J.C., Shoubridge, E.A., Battersby, B.J., 2010. Gimap3 regulates tissue-specific mitochondrial DNA segregation. PLoS Genet. https://doi.org/10.1371/journal.pgen.1001161
Author Response
Response to Reviewer 1 Comments:
Sloane et al.“Mitochondrial DNA Variants and Common Diseases – A Mathematical Model for the Diversity of Age-related mtDNA Mutations. “
Sloane et al.review how mtDNA mutations contribute to a variety of different neurodegenerative diseases, cancer, diabetes, and aging. This revision is much more focused that the previous submission. They provide rationale and mathematical modeling to estimate the number of mtDNA mutations that may accumulate over a person’s lifespan. However, it is unclear what the physiological relevance is to this calculation because it doesn’t reflect mtDNA mutations found in biology (mitotic v. post-mitotic), and the mtDNA replication rate is not a good variable for determining the mtDNA mutational rate in cells (Battersby and Shoubridge, 2001; Jenuth et al., 1997; Jokinen et al., 2010).
Response: We agree that mtDNA mutations are most commonly observed and studied in the context of post-mitotic tissues, and have revised the text to make this point clearer. However, we have also provided citations showing that mtDNA mutations accumulate with age in colonic crypt cells and murine hematopoietic stem cells (PMID: 14597761 and PMID: 17185390), so it would appear that proliferating cells and post-mitotic cells may have alternative ways to accumulate the mutations, which is discussed in the revision. Furthermore, one can still utilize our formula exclusively for post-mitotic by simply ignoring the cell proliferation variable entirely. The main point of this exercise is to stimulate a discussion of the topic.
Regarding the issue of the relationship between mtDNA replication rate and mtDNA mutational rate, that does not seem to be the central issue being addressed in those publications. Also, the premature aging phenotype of the “mutator” mouse with its proof-reading deficient Polg mutation has long been considered fairly strong evidence that mtDNA mutations introduced by PolG are a major player in the aging process (PMID: 15164064).
Also, cell cycle and mtDNA replication are not linked (Bogenhagen and Clayton, 1977).
Response: It is true that mtDNA replication is not directly linked to M phase or some other narrow windows of the cell cycle, but occurs constantly throughout the cell cycle. However, what is also true is that the mtDNA population of a group of cells doubles with every round of cell division. Bogenhagen and Clayton (which was cited by Reviewer 1) stated the following in the first sentence of their abstract: “The number of mitochondrial DNA molecules in a cell population doubles at the same rate as the cell generation time” (Bogenhagen and Clayton, 1977). Thus, for the purposes of our calculation, every time a cell population doubles, the mtDNA population must double as well.
Major Concerns:
1. After the authors have split the original manuscript into two reviews, now this review is lacking a basic introduction to the diseases that it focuses on for the non-expert. It assumes the reader is familiar with all of the neurodegenerative and age-related diseases discussed.
Response: This background was supposed to be provided by the companion review article, “The Role of Mitochondrial-related Nuclear Genes in Age-related Common Disease.” This article was intended to be split off from the original manuscript in consultation with the guest editor for the "Mitochondrial Genetics" issue, Dr. Maria Cristina Kenney, but there still appears to be some disagreement on whether this will be permitted. However, in order to help this manuscript stand better on its own, we have added the additional background to section 2 of the manuscript.
2. In the paragraph starting at lines 350, it unclear why the authors think that the mutant will have a proliferative adaptive during development. If anything, mutant mtDNA (especially mutations in protein coding genes) does not have a proliferative advantage (Fan et al., 2008; Hill et al., 2014).
Response: In most cases, Reviewer 1 is correct that mutant mtDNA would not have a proliferative advantage. However, as we explained in the text, in some cases a mutation can create an advantage of cell proliferation or mtDNA replication. For example, various mutations in the D-Loop could outcompete wildtype mtDNAs while still creating a detrimental effect for its host cell.
Concerns:
1. The grammar and wording again still needs to be rechecked and corrected for this manuscript.
Response: We have gone through the manuscript again to correct the grammar and wording, as requested.
2. Lines 117-118 is not correct. Dementia is a description of symptoms that can be reversible (drug use, vitamin or thyroid deficiency, etc.) or irreversible (AD, FTD, etc.). It is not a potential precursor to AD.
Response: Thank you for pointing this out. You are correct that dementia describes a set of symptoms that occur in a number of diseases, and that AD is only one of those diseases. We have revised this passage accordingly.
3. Lines 186-191 are also not correct. There is not an elevated risk of HD. This is a genetic autosomal dominant disease with either complete or incomplete dominance.
Response: This is correct; the mtDNA variants are downstream of the trinucleotide expansions that are actually associated with HD. We have clarified this in the text.
4. Line 211, define IR. The reviewer thinks you are referring to insulin resistance.
Response: Yes, this was referring to insulin resistance. We have clarified this in the text.
Battersby, B.J., Shoubridge, E. a, 2001. Selection of a mtDNA sequence variant in hepatocytes of heteroplasmic mice is not due to differences in respiratory chain function or efficiency of replication. Hum. Mol. Genet. https://doi.org/10.1093/hmg/10.22.2469
Bogenhagen, D., Clayton, D.A., 1977. Mouse L cell mitochondrial DNA molecules are selected randomly for replication throughout the cell cycle. Cell. https://doi.org/10.1016/0092-8674(77)90286-0
Fan, W., Waymire, K.G., Narula, N., Li, P., Rocher, C., Coskun, P.E., Vannan, M.A., Narula, J., Macgregor, G.R., Wallace, D.C., 2008. A mouse model of mitochondrial disease reveals germline selection against severe mtDNA mutations. Science (80-. ). 319, 958–962. https://doi.org/10.1126/science.1147786
Hill, J.H., Chen, Z., Xu, H., 2014. Selective propagation of functional mitochondrial DNA during oogenesis restricts the transmission of a deleterious mitochondrial variant. Nat. Genet. https://doi.org/10.1038/ng.2920
Jenuth, J.P., Peterson, A.C., Shoubridge, E.A., 1997. Tissue-specific selection for different mtDNA genotypes in heteroplasmic mice. Nat. Genet. https://doi.org/10.1038/ng0597-93
Jokinen, R., Marttinen, P., Sandell, H.K., Manninen, T., Teerenhovi, H., Wai, T., Teoli, D., Loredo-Osti, J.C., Shoubridge, E.A., Battersby, B.J., 2010. Gimap3 regulates tissue-specific mitochondrial DNA segregation. PLoS Genet. https://doi.org/10.1371/journal.pgen.1001161
